# *Snord116*-dependent diurnal rhythm of DNA methylation in mouse cortex

Rochelle L. Coulson[1], Dag H. Yasui [1], Keith W. Dunaway[1], Benjamin I. Laufer [1], Annie Vogel Ciernia[1], Yihui Zhu[1], Charles E. Mordaunt[1], Theresa S. Totah[1] & Janine M. LaSalle [1]

Rhythmic oscillations of physiological processes depend on integrating the circadian clock and diurnal environment. DNA methylation is epigenetically responsive to daily rhythms, as a subset of CpG dinucleotides in brain exhibit diurnal rhythmic methylation. Here, we show a major genetic effect on rhythmic methylation in a mouse *Snord116* deletion model of the imprinted disorder Prader–Willi syndrome (PWS). More than 23,000 diurnally rhythmic CpGs are identified in wild-type cortex, with nearly all lost or phase-shifted in PWS. Circadian dysregulation of a second imprinted *Snord* cluster at the Temple/Kagami-Ogata syndrome locus is observed at the level of methylation, transcription, and chromatin, providing mechanistic evidence of cross-talk. Genes identified by diurnal epigenetic changes in PWS mice overlapped rhythmic and PWS-specific genes in human brain and are enriched for PWS-relevant phenotypes and pathways. These results support the proposed evolutionary relationship between imprinting and sleep, and suggest possible chronotherapy in the treatment of PWS and related disorders.

[1] Medical Microbiology and Immunology, Genome Center, MIND Institute, University of California, Davis, CA 95616, USA. Correspondence and requests for materials should be addressed to J.M.L. (email: jmlasalle@ucdavis.edu)

D aily and seasonal cycles of light, temperature, and feeding govern energy and activity of organisms from all branches of life. These environmental and metabolic inputs play an important role in the synchronization of the core circadian clock with the rhythmic patterns of many physiological and behavioral processes in peripheral tissues[1–3]. The genetically encoded circadian cycle and the environmentally regulated diurnal cycle are integrated by a complex regulatory feedback network which acts at the chromatin, transcriptional, and translational levels to coordinate biological and environmental rhythms[4–6]. In mammals, the core circadian clock resides in the suprachiasmatic nucleus of the hypothalamus; however, almost half of all transcripts, both protein-coding and non-coding, exhibit diurnal rhythms in one or more peripheral tissues[7,8]. The epigenetic mechanisms involved in the diurnal rhythms of the cerebral cortex are less well characterized; however, increasing evidence indicates a role for DNA methylation in these rhythms. In contrast to the historically accepted view of DNA methylation as a stably maintained epigenetic mark, ~6% (25,476) of CpG sites assayed by 450k array are dynamically regulated throughout diurnal and seasonal cycles[9]. This epigenetic plasticity plays an important role in circadian entrainment and the resiliency of the circadian clock to changes in the diurnal environment[10–12]. Studies of shift work have demonstrated that sleep deprivation (SD) results in the epigenetic and transcriptional dysregulation of core circadian genes and increased risk for adverse metabolic phenotypes and cancer[13,14]. The epigenetic and metabolic dysregulation associated with SD in healthy individuals provides a basis for understanding these characteristics in the context of neurodevelopmental disorders, which often share a core sleep phenotype among their unique distinguishing features[15].

Sleep disturbances are common in human neurodevelopmental disorders associated with the 15q11–q13 imprinted locus[15]. This includes Prader–Willi syndrome (PWS) and Angelman syndrome (AS), which are characterized by excessive daytime sleepiness and shorter sleep duration, respectively[16–19]. Genomic imprinting is an epigenetic inheritance pattern that evolved during mammalian radiation concordant with the development of rapid eye movement (REM) sleep, a state of high energy usage in the brain characterized by increased neuronal and metabolic activity and changes in thermoregulation[20]. PWS is characterized by a failure to thrive and hypotonia in early infancy, followed by hyperphagia as well as metabolic, behavioral, cognitive, and sleep abnormalities[16,18,21,22]. Small deletions have defined the minimal PWS critical region to encompass SNORD116[23–26]. Because this locus is maternally imprinted, with expression exclusively from the paternal allele, deletions specifically of paternal SNORD116 result in PWS. The SNORD116 locus includes repeats of small nucleolar RNAs (snoRNAs) processed from a long polycistronic transcript originating at the protein-coding gene SNRPN and extending through non-coding RNAs SNORD116, SNORD115, and the antisense to the maternally expressed UBE3A (UBE3A-AS). The spliced host gene of SNORD116 (116HG) forms a nuclear long non-coding RNA "cloud" that localizes to its site of transcription and is diurnally regulated in size[27,28]. Studies of a PWS mouse model (Snord116[+/−]) carrying a ~150 kb paternal deletion of Snord116 revealed a specific role for 116HG in the diurnal regulation of genes with circadian, metabolic, and epigenetic functions[28,29]. Interestingly, these changes were phase specific, with transcriptional and metabolic dysregulation observed predominately during the light phase[28].

Imprinted genes function cooperatively in a regulatory network to coordinate many biological processes underlying complex phenotypes and molecular mechanisms involved in growth and development, metabolism, and sleep[20,30–32]. Imprinted inheritance patterns of a second imprinted human chromosome locus,

14q32.2, are also observed in Temple and Kagami-Ogata syndromes (TS and KOS). TS and KOS are reciprocally imprinted disorders, with TS caused by maternal uniparental disomy 14 (UPD(14)mat), and KOS caused by paternal uniparental disomy 14 (UPD(14)pat). The 14q32.2 imprinted locus bears striking similarity to the PWS locus, as it encodes the only other repetitive cluster of snoRNAs in the mammalian genome (SNORD113, SNORD114), which are maternally expressed and exhibit allele-specific chromatin decondensation in neurons, similar to SNORD116 and SNORD115[33–35]. TS is caused by the loss of paternal gene expression at this locus, resulting in aberrantly high expression of maternal non-coding RNAs, whereas KOS results from the loss of maternally expressed, non-coding RNAs and the upregulation of paternally expressed DLK1. Interestingly, TS phenocopies PWS, in which paternally expressed non-coding RNAs are lost, suggesting that these two imprinted loci may perform similar functions and share common pathways[36–38]. The loss of Snord116 expression in PWS results in the upregulation of genes in the TS locus in mouse whole cerebral cortex, indicating that the two loci may interact through a cross-regulatory network[28]. In support of this hypothesis, IPW from the PWS locus has been shown to regulate the TS locus in an induced pluripotent stem cell line of PWS[39]. Though both the PWS and TS loci show circadian oscillations in gene expression, the mechanism of this regulation and the impact of circadian rhythms on their cross-regulation are not understood[28,40,41].

Here we investigate the epigenetic rhythms associated with diurnal time and how the loss of 116HG in the Snord116[+/−] mouse model of PWS disrupts the oscillatory pattern of DNA methylation, transcription, and chromatin compaction in the cortex (Fig. 1a). In WT cortex, over 19,000 rhythmically methylated CpG sites reach a nadir (trough) in light hours (ZT6), most of which are lost in the PWS mouse cortex and enriched for human traits associated with body weight and metabolism. Interestingly, the TS locus is enriched for these sites of PWS-disrupted diurnal DNA methylation and exhibits time-disrupted chromatin changes opposite in diurnal time to the light-specific chromatin changes of the PWS locus.

## Results

### Rhythmic DNA methylation is disrupted by loss of Snord116.
Using a PWS mouse model (Snord116[+/−]), we characterized the effect of Snord116 loss on the rhythmicity of the diurnal light cycle to understand the previously described light-phase-specific changes to lipid metabolism and transcription[28]. To identify potential epigenetic alterations in the Snord116[+/−] mice that could explain these diurnal phenotypes, DNA methylation in prefrontal cerebral cortex was assayed in adult (P120-160) males by whole genome bisulfite sequencing (WGBS) and compared to wild-type (WT) littermates (Supplementary Data 1 and Supplementary Fig. 1). Differentially methylated regions (DMRs) were identified based on both time and genotype comparisons for WT and Snord116[+/−] mice (Supplementary Table 1). For time-specific DMRs, zeitgeber time ZT0 (lights on), ZT3 (light), ZT6 (light), ZT9 (light), ZT12 (lights off), and ZT16 (dark) were sequentially compared to each other (i.e., ZT0 versus ZT6 is noted as ZT0v6). Rhythmic DNA methylation sites in cortex were identified as oppositely methylated DMRs at ZT0v6 and ZT6v12, with DMRs identified from ZT0v12 removed (Supplementary Data 2). This analysis revealed a cyclical diurnal pattern for 4,355 DMRs (23,201 CpGs) in WT cortex during light hours, with 3,656 reaching a nadir at ZT6 and 699 reaching a zenith (peak) at ZT6, indicating that DNA methylation levels over these CpG sites fluctuate across diurnal time in mouse cortex (Fig. 1b). The same analyses on the 4,355 WT diurnally cycling DMRs in

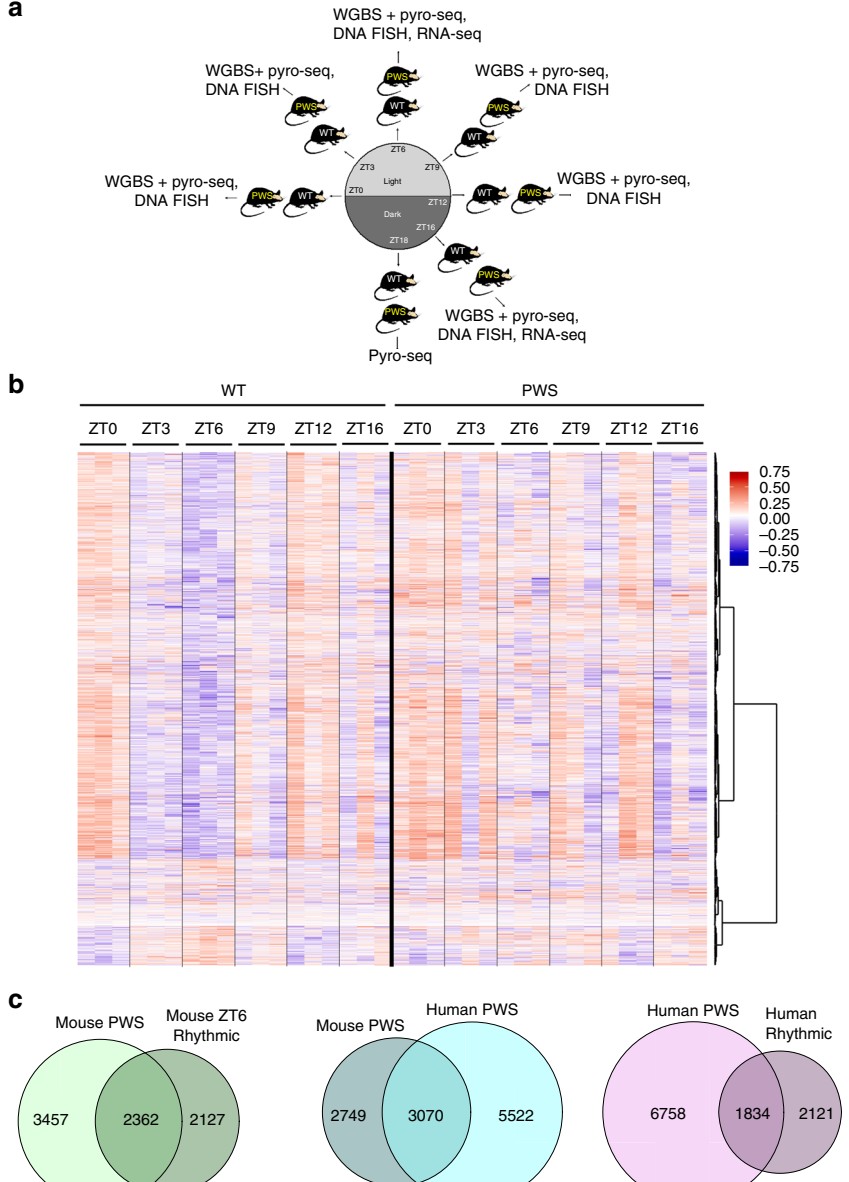

**Fig. 1** Diurnal rhythmic oscillations of DNA methylation are disrupted by loss of *Snord116*. **a** Overview of diurnal experimental design. All experiments were performed on male cerebral cortex. WGBS $n = 3$ WT ZT0, 3 WT ZT3, 3 WT ZT6, 3 WT ZT9, 3 WT ZT12, 3 WT ZT16, 3 PWS ZT0, 3 PWS ZT3, 3 PWS ZT6, 3 PWS ZT9, 3 PWS ZT12, 3 PWS ZT16; RNA-seq $n = 3$ WT ZT6, 2 WT ZT16, 2 PWS ZT6, 2 PWS ZT16; DNA FISH $n = 3$ WT ZT0, 3 WT ZT3, 3 WT ZT6, 3 WT ZT9, 3 WT ZT12, 3 WT ZT16, 3 PWS ZT0, 3 PWS ZT3, 3 PWS ZT6, 3 PWS ZT9, 3 PWS ZT12, 3 PWS ZT16; pyrosequencing $n = 3$ WT ZT0, 3 WT ZT3, 3 WT ZT6, 3 WT ZT9, 3 WT ZT12, 3 WT ZT16, 3 WT ZT18, 3 PWS ZT0, 3 PWS ZT3, 3 PWS ZT6, 3 PWS ZT9, 3 PWS ZT12, 3 PWS ZT16, 3 PWS ZT18. **b** Heat map of WT ZT6 diurnal DMR methylation for WT and PWS cortex. Hyper-methylation (red) and hypo-methylation (blue) respective to the average WT methylation across all diurnal time points. **c** Overlapping genes associated with PWS-specific DMRs and WT rhythmic DMRs in mouse, PWS-specific DMRs in mouse versus human postmortem brain, and human PWS-specific DMR versus human rhythmic from postmortem brain. Methylation data from human postmortem brain was obtained from previously published datasets, GSE81541 (PWS and WT) and https://www.synapse.org/#!Synapse:syn3157275 (Rhythmic)

*Snord116*$^{+/-}$ cortex revealed that only 167 DMRs maintained this rhythmic pattern, with 26 maintaining a ZT6 nadir and 141 maintaining a ZT6 zenith. The oscillation amplitude of rhythmic methylation is greater in WT cortex across diurnal time, with the greatest nadir observed in the early light hours between ZT3 and ZT6 (Supplementary Fig. 2). Overall, 0.35% of potential DMRs (meeting coverage requirements to be considered for DMR calling) in the WT cortex exhibit light-specific rhythmic methylation, with 97% of these disrupted in *Snord116*$^{+/-}$ cortex. Rhythmicity was not a result of low sequence coverage of certain CpGs

(Supplementary Fig. 3a). Additional pyrosequencing analysis of two rhythmic DMRs confirmed the patterns observed by WGBS (Supplementary Fig. 3b).

By combining the DMRs identified between genotypes for each diurnal time point, we identified a set of PWS-specific DMRs in mouse and examined the relationship between methylation changes in PWS-specific genes and genes exhibiting light-specific rhythmicity (Fig. 1c, Supplementary Data 3). 40.6% of genes with PWS-specific methylation changes were rhythmic at ZT6 in WT cortex, which is a significant overlap between the two

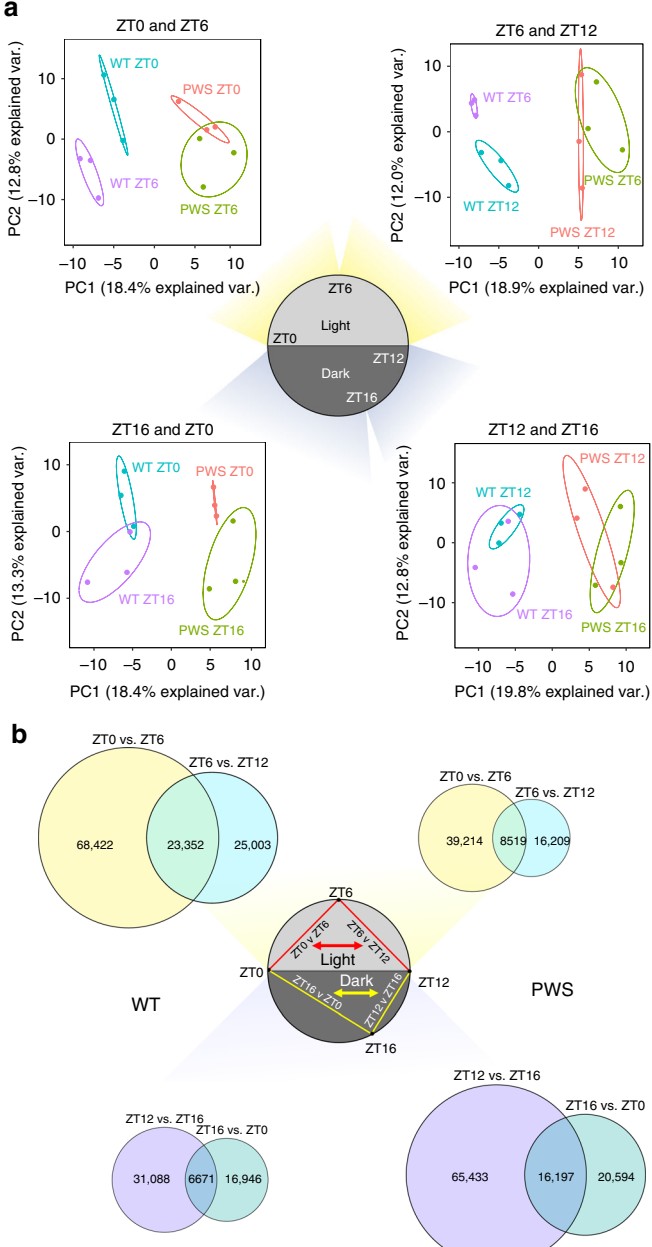

**Fig. 2** DNA methylation is diurnally rhythmic during light hours in WT, but altered to predominantly dark hours in PWS mouse cortex. **a** Principal component analysis of WT vs PWS DMRs. Genotype-specific DMRs separate on the basis of time in WT cortex during light hours, but not in PWS cortex. Ellipses represent the 95% confidence interval for each group. Non-overlapping ellipses indicate a significant difference in methylation profile ($p < 0.05$). **b** Rhythmic CpGs during light and dark hours in WT and PWS cortex. Venn diagram overlaps represent the number of rhythmic CpGs. Top = light hours (ZT0–ZT12), Bottom = dark hours (ZT12–ZT0). 23,201 WT light rhythmic CpGs > 8,417 $Snord116^{+/-}$ light rhythmic CpGs (Fisher's exact test, $p < 0.0001$) and 16,090 $Snord116^{+/-}$ dark rhythmic CpGs > 6,687 WT dark rhythmic CpGs (Fisher's exact test, $p < 0.0001$)

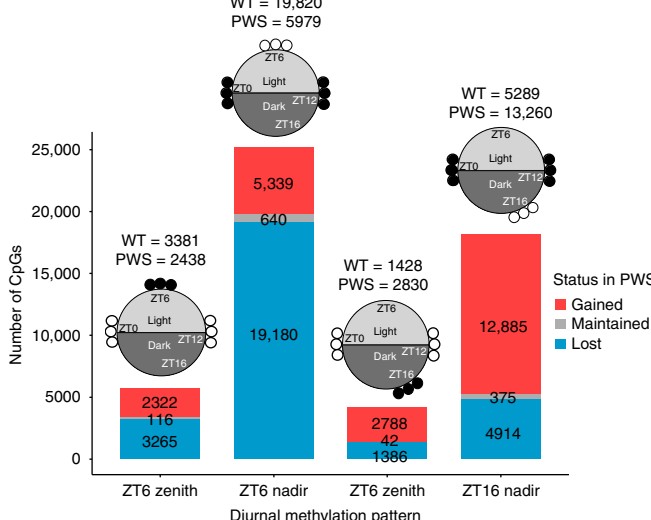

**Fig. 3** The majority of rhythmic CpGs in WT cortex reach a nadir at ZT6, are lost in PWS, and are enriched in promoters and enhancers. Number of diurnally cycling CpGs disrupted in PWS. Diagrams above bars graphically represent how methylation nadirs and zeniths were defined in WT, with open circles representing unmethylated and closed circles representing methylated CpG sites specific to a time point. Total number of CpGs belonging to methylation class indicated above bars

the two species (3,070 genes, $p = 4.7 \times 10^{-221}$, OR = 2.7). We next utilized data from a recent study which evaluated the diurnal and seasonal rhythmicity of DNA methylation in Alzheimer's disease (AD) and control human cortex to determine if genes with dysregulated methylation in human PWS exhibited diurnal rhythmicity[9]. To define a diurnally rhythmic gene set from this study, we empirically required 9% of CpGs assayed for each gene to demonstrate diurnal rhythmicity (Supplementary Fig. 4a). Despite differences in methylation analysis platform (WGBS vs. 450k array), 1,834 diurnally methylated human genes overlapped between studies, representing 21.3% of PWS-specific DMR identified genes ($p = 1 \times 10^{-96}$, OR = 2.1, Fig. 1c and Supplementary Data 3). When PWS DMR genes are filtered for rhythmicity, the significant overlap between mouse and human is maintained, suggesting that rhythmically methylated genes play an essential and conserved role in PWS (250 genes, $p = 3.3 \times 10^{-13}$, OR = 1.7, Supplementary Fig. 4b). These genes are found in pathways involving *Mtor* signaling, circadian entrainment, sleep, diabetes, and cholesterol, including genes from the imprinted KOS/TS locus and the epigenetic enzyme *Tet3* (Supplementary Fig. 4c). Together, these results demonstrate a conservation of PWS-impacted diurnally methylated genes across two mammalian species.

**Rhythmic DNA methylation is dependent on time and genotype.** We next pooled diurnal time points to determine if time could be distinguished as a significant factor in a genotype-specific analysis. For these DMRs, identified solely based on genotype, principal component analysis (PCA) revealed that genotype-specific DMRs could also identify time differences between WT, but not PWS samples for light hours (ZT0–ZT6–ZT12) but not dark hours (ZT12–ZT16–ZT0; Fig. 2a). Due to the uniquely distinct DNA methylation pattern at ZT6, observed by its significant separation from both ZT0 and ZT12 by PCA, we focused our subsequent analyses on this mid light phase time point as an important nadir of DNA methylation within the cortex, with a similar pattern observed at three-hour resolution during light hours, showing that this change in methylation

groups (2,362 genes, $p < 0.0001$, OR = 5.3). In order to assess the relevance of our mouse study to human PWS, we utilized previously published WGBS data to identify human cortex PWS DMRs (Fig. 1c, Supplementary Data 3)[42]. 52.8% of the genes associated with PWS DMRs in mouse overlapped with human PWS-specific DMRs, representing a significant overlap between

**Table 1 Genomic region enrichment analysis of directional diurnal CpGs**

| Whole genome | | | Fold change over expected | | | | Top TF binding sites (total sites) |
|---|---|---|---|---|---|---|---|
| | | | Promoter | Gene body | Enhancers | Intergenic | |
| ZT6 | Nadir | Lost | 0.99[a] | 0.78 | 4.55 | 0.88 | NeuroD1, Nf1, Atoh1, NeuroG2, Tlx (26) |
| | | Gained | 1.96 | 0.76 | 2.39 | 0.86 | NF1, NeuroD1, Ets1, Mef2d, Ebf (6) |
| | Zenith | Lost | 1.41 | 0.88 | 2.59 | 0.83 | X-box, Ap4, MyoD (3) |
| | | Gained | 2.04 | 0.75 | 2.34 | 0.86 | Ctcf, Boris, Rfx, Rfx2 (4) |
| ZT16 | Nadir | Lost | 1.94 | 0.73 | 2.92 | 0.86 | Mef2b, Fra2, NeuroD1, Batf, AP-1 (11) |
| | | Gained | 1.35 | 0.82 | 4.11 | 0.76 | NeuroD1, Atoh1, NeuroG2, Olig2, Nf1 (24) |
| | Zenith | Lost | 2.84 | 0.72 | 1.55 | 0.75 | (0) |
| | | Gained | 1.60 | 0.82 | 2.53 | 0.88 | Brn1 (1) |

Values for fold change over expected are shown. Top five enriched transcription factor binding motifs are listed along with the total number of enriched motifs given in parenthesis (see Supplementary Data 6 for full list)
[a] All categories are enriched or de-enriched at $p$ $5 \times 10^{-5}$ (100,000 permutations), unless noted

begins at ZT3 (Supplementary Fig. 5). These results demonstrate that DNA methylation patterns in the cortex are dependent on both *Snord116* genotype and diurnal time, indicating that analyses of dysregulation in PWS must account for the relationship between these two factors.

To further characterize the individual CpG methylation patterns observed during light hours in WT cortex, we determined which methylation differences between ZT0, ZT6, and ZT12 were rhythmic, defined as reaching either a nadir or zenith at ZT6. Individual CpGs were extracted from each DMR identified between ZT0 and ZT6 or between ZT6 and ZT12 and intersected independently for each genotype (Figs. 2b, 3, and Supplementary Data 4 and 5). During light hours, in which significant time-DMR separations were observed in WT cortex by PCA, the total number of cycling CpGs (either ZT6 nadir or zenith) was significantly greater in WT than *Snord116*[+/−] (23,201 > 8,417, p < 0.0001). The largest represented group of rhythmic CpGs, those that reached their nadir at ZT6, was also significantly greater in WT than in *Snord116*[+/−] (19,820 > 5,979, p < 0.0001) as an isolated group, with nearly all ZT6 CpG nadirs being lost in *Snord116*[+/−] mice. During dark hours, the opposite pattern was observed in which the total number of rhythmic CpGs (either ZT16 nadir or zenith) was significantly greater in *Snord116*[+/−] than WT (16,090 > 6,687, p < 0.0001). Similarly, the largest group of rhythmic CpGs, those that reached their nadir at ZT16, was also significantly greater in *Snord116*[+/−] than in WT (13,260 > 5,289, p < 0.0001), with most ZT16 CpG nadirs being gained specifically in *Snord116*[+/−] cortex. Overall, WT cortex exhibits much greater diurnal rhythmicity in DNA methylation during light hours than PWS cortex; however, this rhythmicity is not completely lost, with a subset of CpGs exhibiting a shift of rhythmicity to the dark phase. Together, these results demonstrate that rhythmic DNA methylation patterns in the cortex are dependent on diurnal time and are disrupted by the loss of *Snord116*.

**Rhythmic DMRs overlap ZT6 transcriptional changes**. To define the functional relevance of CpGs with a dysregulated diurnal cycle in *Snord116*[+/−] mice, we performed a genomic annotation of differentially cycling CpGs by enrichment analysis of promoters, gene body, enhancers, and other intergenic regions. Almost all classes of PWS differentially cycling CpGs were enriched in both promoters and enhancers except the largest class, CpGs that lose their ZT6 nadir in PWS, which were not enriched in promoters (Table 1). The two largest classes of cyclically methylated CpGs, those which lose a ZT6 nadir and those which gain a ZT16 nadir in *Snord116*[+/−] mice, showed a greater than 4-fold enrichment at enhancers, indicating a likely distal regulatory

function. In addition to annotating the location of these CpGs with respect to genes, we performed a transcription factor binding motif analysis on the full DMR regions associated with the cyclically methylated CpGs. DMRs categorized as ZT6 nadir lost and ZT16 nadir gained were enriched in similar transcription factor binding motifs, including several factors involved in neurogenesis and neuronal differentiation (Table 1 and Supplementary Data 6). These include NeuroD1, which regulates both neuronal differentiation as well as insulin expression, and is associated with type 2 diabetes mellitus, an obesity-related complication reported in adults with PWS[43,44]. Although DMRs were not enriched for Bmal1 binding sites, *Snord116*[+/−] genes with dysregulated diurnal expression or methylation include 58.6% of known Bmal1 targets (Supplementary Fig. 6a). Bmal1 interacts with circadian and metabolic pathways such as Mtor/Akt, which is known to be diurnally dysregulated by *Snord116* loss[28].

24.7% of genes that lost a ZT6 nadir overlapped those that gained a ZT16 nadir, indicating a nadir shift from light to dark hours in PWS (Fig. 4a). We previously showed the loss of *Snord116* resulted in the increased expression of 6,067 genes specifically at ZT6 in cortex, including genes involved in circadian rhythms, energy expenditure, and DNA methyltransferase activity (Supplementary Data 7)[28]. To examine the relationship between *Snord116*[+/−] dysregulated transcripts and genes associated with altered diurnal DMRs, we calculated the percentage of genes upregulated at ZT6 that were also associated with each class of PWS-specific diurnal DMRs. Figure 4b shows that 23.3% of genes upregulated at ZT6 in *Snord116*[+/−] cortex also lost a methylation nadir at ZT6 and 20.7% gained a methylation nadir at ZT16. Overall, the degree of overlap with DMRs and genes upregulated at ZT6 showed a reciprocal pattern between the light and dark hours. Interestingly, genes with altered diurnal methylation at ZT16 showed a greater overlap with genes upregulated at ZT6 than with genes upregulated at ZT16, providing evidence for a time-lagged relationship between methylation and expression changes of *Snord116* altered genes. Of the genes that were upregulated at ZT6, 9.6% also shifted their methylation nadir from ZT6 to ZT16 (Fig. 4b and Supplementary Fig. 6b). Enrichment for enhancers and promoters within these overlapped subsets of genes was similar to that observed for the whole genome, with PWS ZT6 nadir lost CpGs becoming significantly de-enriched in the promoters of ZT6 upregulated genes (Supplementary Table 2). As seen for the whole genome, CpGs with a ZT6 nadir lost or ZT16 nadir gained in PWS showed the greatest enrichment in enhancers with a greater than 3-fold enrichment. Overall, these data indicate that the most prevalent disruptions in rhythmic DNA methylation in *Snord116*[+/−] cortex are ZT6 nadirs lost and ZT16 nadirs gained in PWS,

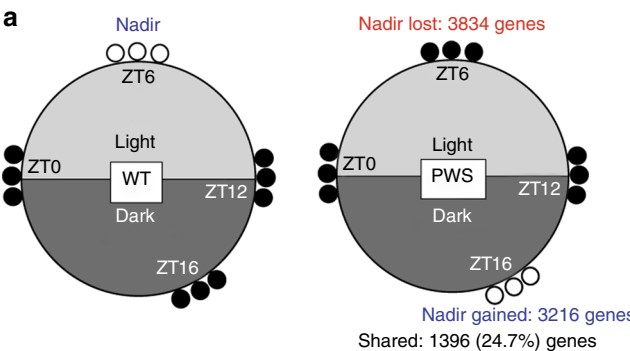

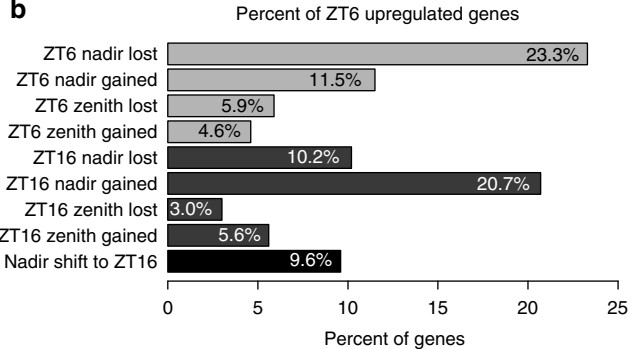

**Fig. 4** Genes which lose a ZT6 methylation nadir or gain a ZT16 nadir in PWS overlap significantly with genes upregulated at ZT6 and enhancers. **a** Graphical representation of the rhythmic DNA methylation pattern changes in PWS for the two largest groups of genes (ZT6 nadir lost and ZT16 nadir gained). Filled circles represent methylated and empty circles represent unmethylated CpGs. **b** Percent overlap of genes upregulated at ZT6 in PWS mouse cortex by RNA-seq analysis with each PWS-specific diurnal methylation category

corresponding with genes showing increased expression at ZT6 ($p = 3.54 \times 10^{-74}$ and $p = 5.49 \times 10^{-83}$, respectively) and transcription factor binding sites in promoters and enhancers. Changes in diurnal expression are therefore associated with shifted diurnal methylation across light and dark periods, potentially through rhythmic transcription factor binding at enhancers.

**Disrupted rhythmic DMR genes are enriched for PWS pathways.** For the genes associated with ZT6 nadir gain (enriched in promoters and enhancers) that were also upregulated at ZT6 in $Snord116^{+/-}$ cortex, we performed an analysis of gene functions and human phenotypes (Fig. 5 and Supplementary Data 8). KEGG pathway analysis revealed that these $Snord116^{+/-}$ epigenetically dysregulated genes were significantly enriched for functions in circadian entrainment, AMPK signaling, stem cell pluripotency, axon guidance, insulin resistance, and other functions related to addiction, cancer, and calcium signaling. Within the database of Genotypes and Phenotypes (dbGaP), these genes were enriched for many phenotypes relating to body mass index (BMI), cholesterol, waist circumference, triglycerides, and other PWS-relevant phenotypes. These analyses provide support for the relevance of the disruption of the diurnally rhythmic transcriptome and epigenome in the pathology of PWS, integrating the sleep and metabolic phenotypes observed in these patients.

**Another imprinted snoRNA cluster is regulated by $Snord116$.** An emerging "genomic imprinting hypothesis of sleep" proposes a connection between the evolution of parent-of-origin

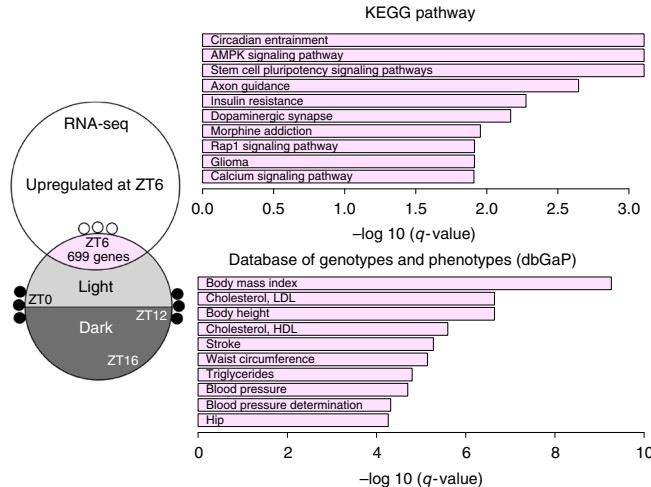

**Fig. 5** Genes with dysregulated DNA methylation and expression at ZT6 are enriched for human PWS-relevant phenotypes and pathways. Filled circles represent methylated and empty circles represent unmethylated CpGs. Top ten terms graphed in bar plots as $-\log_{10}(q\text{-value})$ (Fisher's exact test with Benjamini–Hochberg correction). See Supplementary Data 8 for full list of terms

inheritance and the development of rapid eye movement (REM) sleep patterns, although the direct mechanism is unknown[20]. To explore the potential co-regulation of imprinted loci with respect to the diurnal cycle, we examined the enrichment of time- and $Snord116$-specific differentially methylated CpGs within imprinted gene clusters (Supplementary Data 9). Enrichment within specific imprinted loci was largely unique to CpGs with a ZT6 nadir loss in $Snord116^{+/-}$ mice, although no overall enrichment for imprinted loci was observed. The $Begain$-$Dio3$ cluster of imprinted genes within the TS locus, which is the only other snoRNA-containing gene cluster in the genome, showed the greatest enrichment for $Snord116$ ZT6 nadir lost CpGs, with a greater than 2-fold enrichment over expected ($p = 1.33 \times 10^{-4}$; Fig. 6a). These 45 CpGs are clustered into six DMRs, which contain binding sites for transcription factors such as NeuroD1, NeuroG2, Foxo1, Ctcf, Boris, Npas, and Bmal1 (Supplementary Data 10). In addition to altered diurnal methylation, expression of maternal genes within the TS locus are significantly upregulated at ZT6 in $Snord116^{+/-}$ mice (Fig. 6b)[28].

To further characterize the diurnal mechanism of rhythmic methylation and expression between these two imprinted loci, we performed DNA fluorescence in situ hybridization (FISH) to examine chromatin dynamics over time, since both snoRNA-containing loci undergo chromatin decondensation of only the active allele specifically in neurons[34]. Chromatin decondensation of the PWS paternal allele and TS maternal allele, as well as the distance between the two was measured in WT and $Snord116^{+/-}$ cortical neurons at ZT0, ZT3, ZT6, ZT9, ZT12, and ZT16 (Fig. 7 and Supplementary Data 11). In WT neurons, the PWS locus exhibited rhythmic decondensation during light hours between ZT3 and ZT12, with a nadir at ZT6. Loss of $Snord116$ resulted in significantly more condensed paternal chromatin during the light hours; however, this significant genotype effect was absent at ZT16, during the dark phase. The loss of $Snord116$ also corresponded to a progressive lengthening of the TS locus, resulting in a significantly longer decondensed TS maternal allele at ZT12 and ZT16 in $Snord116^{+/-}$ cortical nuclei ($p = 8.1 \times 10^{-5}$ (0.2 μm) and $4.4 \times 10^{-2}$ (0.14 μm), respectively).

Chromatin isolation by RNA purification (ChIRP) analysis, in which the $116HG$ and chromatin to which it is bound was

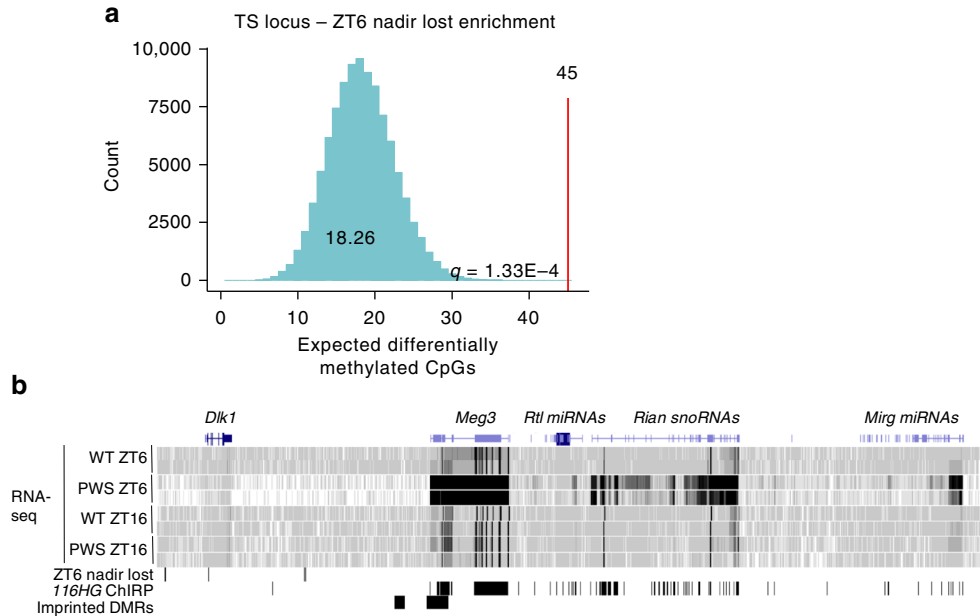

**Fig. 6** The imprinted TS locus is enriched for dysregulated diurnal methylation and expression in *Snord116*[+/−] cortex. **a** CpGs which lose their DNA methylation nadir at ZT6 with the loss of *Snord116* show the greatest enrichment at the TS locus compared to all imprinted loci (Fisher's exact test with Benjamini–Hochberg correction, $q = 1.33 \times 10^{-4}$). (See Supplementary Data 9 for full imprinted loci enrichment analysis). **b** UCSC Genome Browser view of the expression profile and *116HG* ChIRP binding peaks across the TS locus. The PWS *116HG* RNA cloud colocalizes with the TS locus and loss of *Snord116* results in the upregulation of TS locus genes specifically during light hours (ZT6)

isolated, indicates that the TS and PWS loci colocalize within the *116HG* RNA cloud formed by the PWS locus; however, DNA FISH analysis revealed heterogeneity in the distance between the two loci for each neuron, possibly reflecting an unsynchronized interaction, with only a fraction of neurons exhibiting colocalization at any given time (Fig. 6b and Supplementary Fig. 7)[28]. The interaction between the two loci indicated by ChIRP was mediated by the *116HG* RNA cloud, which is diurnally dynamic in size and reaches a diameter of ~1.5 μm at ZT6[28]. Furthermore, several genes encoding epigenetic factors involved in the deposition, removal, or recognition of DNA methylation were identified as either localized within the ZT6 *116HG* RNA cloud by ChIRP or transcriptionally upregulated by RNA-seq in *Snord116*[+/−] (Table 2). Together, these results suggest that the time-dependent chromatin dynamics of the PWS locus DNA and the *116HG* RNA mediate the genome-wide and TS locus-specific changes to DNA methylation in a time-dependent manner (Fig. 8).

## Discussion

This study surveyed PWS across seven diurnal time points, integrating genome-wide sequencing with targeted molecular approaches, resulting in multiple novel insights into the relationship between circadian rhythms, epigenetics, and two human imprinting disorders. First, we demonstrated the rhythmic pattern of the cortical methylome and its disruption in a genetic mouse model of PWS, showing conservation of rhythmic methylation across two mammalian species. Second, we identified genes significantly enriched for PWS-relevant phenotypes of body mass index (BMI) and circadian-entrained metabolic pathways from the genes that lost rhythmic DNA methylation and gained expression in PWS mice. Finally, we identified the only other imprinted snoRNA cluster in the mammalian genome as reciprocally altered in diurnal chromatin dynamics, providing a potential mechanism for the time-shift of DNA methylation patterns.

Unique evolutionary pressures between mouse and human have introduced important differences between the species which must be considered in the translation of findings from mouse studies to humans[45,46]. Despite differences in size and synaptic complexity between the mouse and human cerebral cortex, its role in cognitive and executive function is common between species. Due to the nocturnal rhythm of mice and differences in PWS phenotype between mice and humans, we evaluated the significance of our mouse study using comparative analyses across species for phenotype and rhythmicity. Significant overlap between mouse and human PWS differentially methylated genes demonstrates the relevance of *Snord116*[+/−] mice in modeling human PWS. In addition, PWS differentially methylated genes in both species significantly overlapped with previously described rhythmically methylated genes[11], despite genomic coverage differences in assays. The light-specific rhythmic CpGs we observed in mouse comprised 0.39% of all CpGs meeting the criteria for DMR consideration during light hours, which is significantly greater than expected by chance (0.034%) and significantly greater than PWS rhythmicity (0.11%) (Fisher's exact test $p < 0.0001$ for both comparisons, Supplementary Table 3). These early light phase nadirs in DNA methylation significantly contribute to the changing methylation patterns across diurnal time as identified by PCA, however they do not represent the exclusive nadir of DNA methylation for all potentially rhythmic CpGs within the genome. A recent large-scale study of human neo-cortex DNA methylation reported diurnal and seasonal rhythmicity in 6% of all CpGs assayed by 450k array ($p < 0.05$, uncorrected)[11]. The difference in the proportion of rhythmically methylated regions between our studies is most likely due to the 450k array representing a limited subset of CpGs that are preferentially selected to be associated with known promoters and enhancers, regions that we found to be preferentially enriched for rhythmic methylation. Due to the light phase specific transcriptional and metabolic dysregulation demonstrated previously in this PWS mouse model[28], we sampled the dark phase at a much lower density than the light phase, thereby limiting the potential

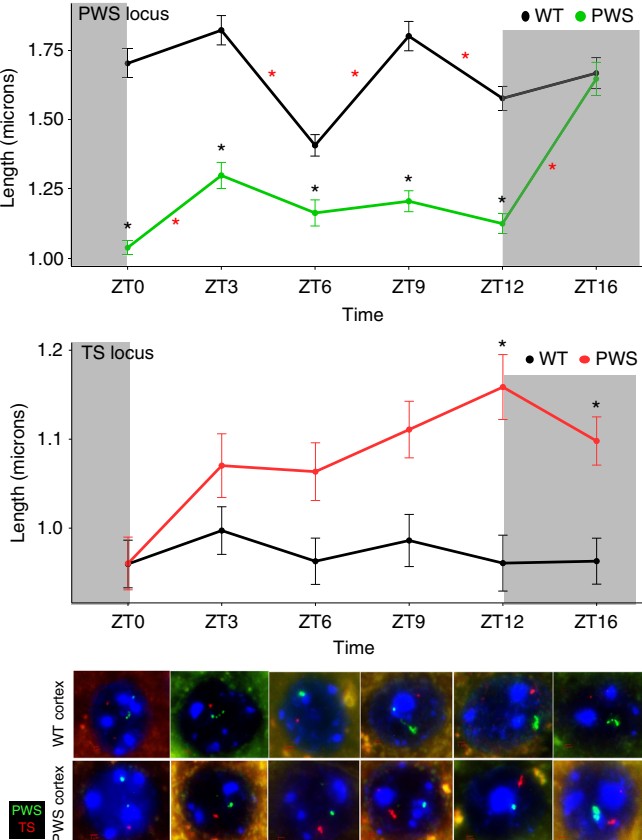

**Fig. 7** *Snord116* modifies chromatin decondensation of both *Snord* gene clusters in a time-dependent manner. DNA FISH for the diurnal decondensation of the PWS (green) and TS (red) loci, including representative images for each condition. Compact silent alleles of each locus are much smaller and fainter than decondensed active alleles. Exposure was optimized to the bright decondensed alleles and decondensation of the largest allele for each locus was measured. Scale bars represent 1 μm. *Snord116*[+/−] neurons exhibit light-specific loss of chromatin rhythmicity at the PWS locus and dysregulation of the TS locus. $N = 300$ nuclei from 3 cortices for each genotype and time point, error bars represent s.e.m. Red asterisks indicate significance between time points within the genotype and black asterisks indicate significance between genotypes for a particular time point. Mixed model ANOVA with Tukey correction, see Supplementary Data 11 for full list of *p*-values

**Table 2 Several epigenetic factors are upregulated at ZT6, likely playing a role in the dysregulation of rhythmic DNA methylation**

| ZT6 upregulated epigenetic factors | |
| --- | --- |
| *Dnmt1* | *Tet1* |
| *Dnmt3a** | *Tet2* |
| *Mbd1* | *Tet3* |
| *Mbd4* | *Mecp2** |
| *Mbd5* | *Arntl** |
| *Hdac3* | *Npas2* |
| *Hdac4* | *Neurod1* |
| *Hdac5* | |

* Genes directly interact with the *116HG* RNA cloud

identification of DNA methylation nadirs and zeniths during the dark phase. Future studies with increased dark phase sampling could improve the resolution to distinguish between loss and phase shift of rhythmicity in PWS cortex. Despite these limitations, the concordance between human and mouse in these comparative analyses indicate strong biological significance of rhythmic DNA methylation in the human brain and specifically in the pathogenesis of PWS.

Remarkably, the intersection of methylation and transcriptional data revealed a set of diurnally dysregulated genes that were enriched for human traits of BMI, cholesterol, triglycerides, and waist circumference, which are all important in the obesity phenotypes in PWS as well as for the emerging link to meal timing and circadian biology in common human obesity[1,47]. Pathways enriched for PWS diurnally altered genes were also enriched for functions involving circadian entrainment, AMPK signaling pathways, insulin resistance, as well as axon guidance and dopaminergic synapses relevant to the neurodevelopmental abnormalities in PWS. These new analyses of DNA methylation

in the context of both *Snord116* genotype and diurnal time have refined our previous description of transcriptional alterations to *Mtor* and diurnal metabolism to pinpoint the genes most critical for the PWS phenotype.

The apparent mechanism of diurnal methylation changes to 0.39% of CpG sites involves a combination of diurnally dynamic chromatin accessibility of two imprinted snoRNA clusters and the epigenetic factors mediating the changes that are also regulated by circadian rhythms. Interestingly, loss of *Snord116* results in the ZT6 upregulation of *Dnmt3a* and *Dnmt1*, potentially mimicking the defect in diurnal synchronization observed in sleep deprivation studies[28]. Our results are consistent with aberrantly high levels of DNA methyltransferases associated with sleep deprivation, in which epigenetic, transcriptional, and metabolic dysregulation has been observed[13,48]. Genes upregulated in ZT6 *Snord116*[+/−] cortex showed loss of rhythmic methylation at ZT16, suggesting a time-delayed relationship between rhythms of transcription and DNA methylation in diurnal time. Aberrantly high expression of epigenetic and circadian regulators at ZT6 may lead to the prolonged accumulation of these proteins into the dark phase, resulting in the shifted methylation changes observed in PWS. The rhythmicity of epigenetic enzymes plays a role in many biological processes in a diverse array of organisms. Circadian expression of epigenetic factors including DNMTs, TETs, MBDs, and HDACs have been observed in the central circadian oscillator of the SCN as well as other metabolically important peripheral tissues such as liver and muscle, regulating processes including lipid metabolism, seasonal reproductive rhythms, and torpor-arousal bouts during winter hibernation[10,49–53]. Through its own rhythmic chromatin decondensation and dynamic RNA cloud, *Snord116* may act to orchestrate the rhythmic patterns of some of these epigenetic factors within the brain.

Unlike the TS locus, in which chromosomal arrangement is conserved between marsupials and eutherians, the genomic organization of the PWS locus is unique to eutherian mammals[54,55]. Interestingly, despite the earlier evolutionary origin of the TS locus, acquisition of imprinting control regions and subsequent establishment of imprinting status at these two loci evolved concurrently within the eutherian lineage[54,56]. The gain of imprinting at the TS locus concomitant with the assembly of the PWS locus supports the hypothesis of a cross-regulatory network between the two loci. In the absence of the *116HG*, the TS locus, carrying the only other snoRNA cluster in the genome, loses rhythmic methylation marks and may decondense to compensate for PWS locus compaction in a diurnally regulated manner.

Understanding the diurnal timing of dysregulation provides important insights into the molecular pathogenesis of PWS and the best timing for therapies. The majority of commonly used

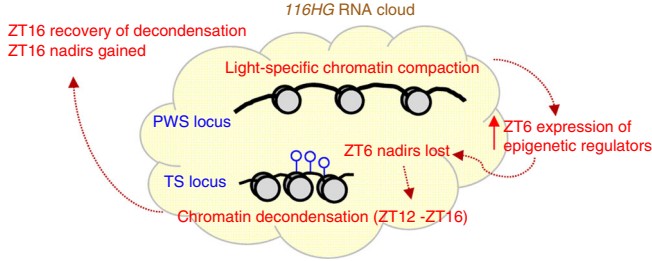

*116HG* RNA cloud

ZT16 recovery of decondensation
ZT16 nadirs gained

Light-specific chromatin compaction

PWS locus

ZT6 expression of epigenetic regulators

ZT6 nadirs lost

TS locus

Chromatin decondensation (ZT12 -ZT16)

(Red text indicates changes observed in PWS)

**Fig. 8** Model of dysregulated epigenetic events throughout the diurnal cycle in PWS. The PWS locus becomes condensed during light hours (ZT0–ZT12). Epigenetic regulators are expressed at aberrantly high levels at ZT6 and DNA methylation nadirs are lost at ZT6 (including at the TS locus). The TS locus undergoes delayed decondensation (ZT12–ZT16), potentially to compensate for PWS locus compaction. By ZT16, PWS decondensation is restored and DNA methylation nadirs are gained, representing a diurnal shift in rhythmic methylation and a light-specific dysregulation of chromatin decondensation. Red text indicates changes observed in *Snord116$^{+/-}$*. Open circles indicate unmethylated CpGs

drugs in the United States target diurnally oscillating genes (56 of the top 100)[7]. The short half-life of many of these drugs and the rhythmic expression of their targets suggests that the time of administration plays a potentially important role in the efficacy of treatment. By characterizing the epigenetic mechanisms of PWS with respect to the diurnal cycle, this study demonstrates the potential application of chronotherapy for the treatment of PWS as well as other epigenetically regulated neurodevelopmental, metabolic, and sleep disorders.

## Methods

**Whole genome bisulfite sequencing (WGBS)**. Mice were sacrificed at ZT0, 3, 6, 9, 12, and 16. Genomic DNA was isolated from 5 to 10 mg prefrontal cortex using the Gentra Puregene kit (Qiagen) and 500ng DNA was bisulfite converted using the EZ DNA Methylation-Lightening kit (Zymo). Libraries were prepared with 100ng bisulfite converted DNA using the TruSeq DNA Methylation kit (Illumina) and barcoded using TruSeq DNA Methylation Kit Index PCR Primers (Illumina). Libraries were sequenced on the HiSeq4000 at the Vincent J. Coates Genomics Sequencing Laboratory at UC Berkeley. Reads were aligned as described previously[42]. A mapping stringency of three mismatches for reads without adapters (90 bp after quality trimming) and two mismatches for adapter trimmed reads (80 bp after quality and adapter trimming) was used for alignment by BSseeker2.

**Differentially methylated regions (DMRs)**. DMRs were called as described previously[42]. To call a DMR, all samples must have at least one read in each CpG, each DMR must contain a minimum of three CpGs, each CpG must be within 300 bp of the previous CpG to be included in the same DMR, the smoothed methylation difference between the two groups must be at least 10%, and the unadjusted p-value for each CpG must be less than 0.05 to be included in a DMR.

**Determination of human PWS DMRs**. DMRs were called as in mouse using previously published WGBS data to identify human cortex PWS DMRs (GEO GSE81541)[42]. Cortical tissue was sampled from BA9 (prefrontal cortex) and BA19 (temporal cortex), which are regions implicated in the cognitive deficits of PWS and the phenotypically related TS.

**Assignment of ZT6 diurnally rhythmic DMRs and CpGs**. Total DMRs and CpGs assayed during DMR analysis were intersected for all samples during light and dark hours to ensure that all sites analyzed for diurnal cycling were represented in each sample. To identify diurnal rhythmicity during light hours, DMRs and CpGs that were called as differentially methylated for both the ZT0–ZT6 and ZT6–ZT12 comparisons were examined for both WT and *Snord116$^{+/-}$* mice. If a site was hypomethylated between ZT0 and ZT6 and hypermethylated between ZT6 and ZT12, it was called diurnally cycling, specifically with a nadir at ZT6. DMRs and CpGs with the opposite pattern were identified as having a zenith at ZT6. For increased stringency, any sites called as a DMR between ZT0 and ZT12 were removed, as methylation at these sites did not return to baseline during the cycle. All DMRs and CpGs were intersected with shared total assayed sites to ensure that coverage requirements were met for all samples. A background set of all covered genomic locations assayed for DMRs was used for normalization. This analysis was

repeated for cycling during dark hours, analyzing differential methylation between ZT12–ZT16 and ZT16–ZT0 comparisons for each genotype. WT and *Snord116$^{+/-}$* sites were intersected to identify diurnal rhythmicity that was lost, gained, and maintained across genotypes during light and dark hours. A null distribution for rhythmicity was created by permutation (3× per genotype) of light hour ZTs used for original rhythmicity analysis. Biological replicates of ZT0, ZT6, and ZT12 were shuffled and reassigned to each time point. DMRs were then called for each permutation as described previously and rhythmicity analysis was performed for each permutation as described for experimental groups. Permutations were averaged to obtain a null expected rhythmicity for each genotype and rhythmicity was tested against the null distribution for each genotype as well as between WT and PWS by Fisher's exact test. Results of permutation tests are shown in Supplementary Table 3.

**Hierarchical clustering**. Methylation was called for every sample across each DMR identified as diurnally rhythmic during ZT6 in WT cortex using a custom roi script (https://github.com/kwdunaway/WGBS_Tools—roi). Percent methylation of each sample was normalized to the mean methylation of all diurnal time points in WT cortex. DMRs were sorted using Ward's method of hierarchical clustering.

**Oscillation amplitude analysis**. Using methylation values from hierarchical clustering, z-scores were calculated by scaling methylation across all samples for each rhythmic DMR. Histograms of z-scores were graphed as genotype comparisons for each time point as well as time point comparisons for each genotype. Z-scores represent the number of standard deviations from the mean of all conditions, with larger absolute values indicating greater deviation from the mean and a larger oscillation amplitude.

**Rhythmic coverage analysis**. Methylation and coverage for each sample across all DMRs identified as diurnally rhythmic during ZT6 in WT cortex was called using a custom roi script (https://github.com/kwdunaway/WGBS_Tools - roi). DMRs were binned by average coverage across all samples and the methylation for each CpG at each coverage level was graphed for the original time points used to call rhythmicity (ZT0, ZT6, and ZT12).

**Identification of human rhythmic genes**. The human data was obtained from Lim et al.[9], where CpG methylation in the dorsolateral prefrontal cortex was examined using the Infinium HumanMethylation450 BeadChip (Illumina). The processed data contained 420,132 probes belonging to autosomal CpG sites that passed quality control. This data was filtered for significant (nominal p < 0.05) diurnal rhythmicity, which left 25,476 CpG sites. The CpG IDs were mapped to hg19 using the FDb.InfiniumMethylation.hg19 Bioconductor package (http://bioconductor.org/packages/FDb.InfiniumMethylation.hg19/). The genomic coordinates were then annotated to genes using the Genomic Regions Enrichment of Annotations Tool (GREAT)[57].

**Pyrosequencing**. DNA was isolated and bisulfite converted as for WGBS. PCR and sequencing primers were designed using the PyroMark Assay Design Software 2.0 (Qiagen). DMRs were amplified using the PyroMark PCR kit (Qiagen) according to the manufacturer protocol. Pyrosequencing was performed using the PyroMark CpG Software 1.0.11 (Qiagen) on the PSQ 96MA (Biotage). 4 µl of Streptavidin Sepharose High Performance beads (GE Healthcare) and 36 µl of PyroMark Binding Buffer (Qiagen) were added to each PyroMark PCR product and shaken for 10 min. 2 µl of 10 µM sequencing primer was added to 38 µl PyroMark Annealing Buffer (Qiagen). Vacuum probes were washed twice with ddH$_2$O, then placed in the PCR product bead mix to capture target regions. Probes were then placed sequentially in 70% ethanol, PyroMark Denaturation Buffer (Qiagen), and 1× PyroMark Wash Buffer (Qiagen). Vacuum was removed and beads were washed from probes in the sequencing primer mix. The plate was heated at 80 °C for 2 min, then cooled to room temperature. Enzyme, Substrate, and dNTPs (PyroMark Gold Q96 Reagents, Qiagen) were added to the PyroMark Q96 Cartridge 0007 (Qiagen) according to PyroMark CpG Software 1.0.11 specifications for each assay design. Three biological replicates were sequenced in duplicate for each genotype and time condition. Methylation was averaged across CpGs for each DMR and DMR methylation for each condition was graphed with S.E.M.

**Chr 17 DMR primers**. TGAGGAAAGGGTTTGGAAGGTTAT (Forward)
TCCAAAAACATCATACAACCTATATCTAA (Reverse, 5′ Biotin)
TGGAAGGTTATTTGTAGT (Sequencing)

**Chr 5 DMR primers**. AAGTTGAGTTGGAGTAGTGG (Forward)
CCCAACTTTCCTTTCACC (Reverse, 5′ Biotin)
ATTGAAGTGTATTTGTTAGG (Sequencing)

**Principal component analysis (PCA)**. All diurnal time points were pooled for each genotype and DMRs were called on the basis of genotype (WT vs. PWS). Percent methylation was calculated over each DMR for every genotype-time

condition using a custom roi script (https://github.com/kwdunaway/WGBS_Tools#roi). To identify time-specific differences in methylation, Principal Component Analysis was performed on DMRs across diurnal time points using the prcomp function and ggbiplot package in R. The 95% confidence interval was plotted as ellipses for each group. Non-overlapping ellipses indicate a statistically significantly difference in methylation pattern between groups ($p < 0.05$).

**Assignment of DMRs to enhancers**. Enhancers were assigned using data from the Ren lab (UCSD) Mouse Encode Project (http://chromosome.sdsc.edu/mouse/download.html). For all enhancers, chromatin signature data for mouse cortex was used. Because enhancers are often difficult to assign to the nearest gene due to their long-range effects, enhancers believed to regulate the expression of the ZT6 upregulated genes were selected using cortex-specific enhancer–promoter units (EPUs), which were filtered for EPUs involving genes upregulated at ZT6.

**Enrichment analysis**. Enrichment of diurnally rhythmic CpGs within genomic regions (promoter, gene body, enhancers, intergenic) and imprinted loci was assayed using GAT—Genomic Association Tester (https://github.com/AndreasHeger/gat). The workspace (effective genome) included all CpGs assayed by the DMR finder, which requires coverage of at least one read in all samples and three CpGs with ≤300 bp between each site. Annotation files contained all regions to be tested for enrichment, which were analyzed together and corrected for multiple hypothesis testing (Benjamini–Hochberg). Isochores were included to account for variable GC composition and 100,000 iterations were used for permutation testing.

**Transcription factor binding enrichment**. DMRs were examined for enrichment of known transcription factor binding motifs using Hypergeometric Optimization of Motif EnRichment (HOMER) (http://homer.ucsd.edu/homer/index.html). All possible DMRs assayed, chopped to the size distribution of significant DMRs, were used as custom background regions and percent CpG content was used for normalization.

**Pathway and phenotype enrichment**. Genes were assigned to DMRs and CpGs using the Genomic Regions Enrichment of Annotations Tool (GREAT)[57] with the default association settings. Genes with upregulated expression at ZT6 were intersected with genes with altered DNA methylation rhythmicity and the shared genes were analyzed for enrichment using Enrichr[58,59], with terms sorted for adjusted p-value.

**DNA FISH and microscopy**. BACs were ordered from BACPAC Resources (Children's Hospital Oakland Research Institute). PWS locus BACs: RP24-275J20, RP23-358G20, RP23-410L2, RP24-147O20, RP23-141P24. TS locus BAC: RP23-60E10. Nick translation of DIG and biotin labeled probes and DNA FISH were performed as described previously[34]. Slides were imaged on an Axioplan 2 fluorescence microscope (Carl Zeiss) equipped with a Qimaging Retiga EXi high-speed uncooled digital camera and analyzed using iVision software (BioVision Technologies). Images were captured using a ×100 oil objective and ×1 camera zoom with blue, red, and green filters at 0.1 μm sections and z-sections were stacked. PWS and TS locus decondensation was measured as the distance along the appropriate fluorescent signal. The compact allele at each locus is much smaller and dimmer than the decondensed allele, and all measurements were taken for the active, decondensed allele. Exposure time was optimized for the intensity of the bright decondensed alleles. The distance between the two loci was measured as the shortest distance between the two loci divided by the nuclear diameter in the same plane. All measurements were converted from pixel counts to microns according to objective and zoom (1px = 0.069 μm). All measurements were blinded to minimize bias.

**Research animals**. All experiments were conducted in accordance with the National Institutes of Health Guidelines for the Care and Use of Laboratory Animals. All procedures were approved by the Institutional Animal Care and Use Committee of the University of California, Davis.

**Code availability**. Custom scripts for WGBS analysis are available at https://github.com/kwdunaway/WGBS_Tools. Custom scripts for DMR calling are available at https://github.com/cemordaunt/DMRfinder.

**Data availability**. Whole genome bisulfite sequencing data that support the findings of this study have been deposited in the Gene Expression Omnibus (GEO) with the accession number GSE103249. Figures 1–6 were produced using these data.

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

## Acknowledgements

Work funded by National Institutes of Health 5T32GM007377-35 (R.L.C.), U54 HD079125, R56NS076263, and 1R01NS076263 (J.M.L.) All animal procedures are covered by the UC Davis IACUC protocols #18120 and #19075 This work used the Vincent J. Coates Genomics Sequencing Laboratory at UC Berkeley, supported by NIH S10 OD018174 Instrumentation Grant.

## Author contributions

R.L.C. and J.M.L. designed experiments and wrote manuscript. R.L.C. and Y.Z. performed experiments. R.L.C., B.I.L., and A.V.C. performed data analysis. K.W.D. and C.E.M. provided tools for WGBS analysis. R.L.C., D.H.Y., and T.S.T. performed tissue dissection and mouse husbandry.

## Additional information

**Competing interests:** The authors declare no competing interests.

