## [Peer Review File(PDF 278 kb) · Nature Communications]

Reviewers' comments:

Reviewer #1 (Remarks to the Author):

Review NCOMMS-17-23671-T

The manuscript by Coulson et al., has identified robust circadian oscillations in DNA methylation within the mouse cortex. These findings are exciting and have significant implications for our understanding of basic molecular signaling pathways, and the potential for short-term epigenetic medications in a complex neural tissue. The manuscript is largely a bioinformatic study that has examined DNA methylation of cortical genome in wild-type and heterozygous snord116 mice. The DNA methylation patterns are then compared with previously published human tissue, which afford the ability to identify specific-specific as well as potential translatable results. The comments below are for the author's consideration and should not reduce the excitement for the data presented in the manuscript.

1) The manuscript cannot be properly evaluated at this time because not all the material was provided in the attachments available online. Specifically, the supplementary material is not organized and it is difficult to locate the specified files. For example, the 'Supplementary Table' attachment only has Table S12; and does not have Table S1-S11. I could not locate where these Tables were presented. There were four Supplementary Figures (Figure S1-S4) attachments available, but these are not provided in the main text. This could easily be rectified in a revised manuscript.

2) The overall story lacks specificity and instead is presented in a half-hazard manner. The mouse model selected is suggested to be a model for Prader-willis syndrome, the authors then emphasize a genomic region implicated in Temple/Kagami-Ogata syndrome and also discuss the findings related to Prader-willis and energy balance. The manuscript would significantly benefit from a streamlined approach; less on the 'translational' aspects and more on the exciting findings on basic mechanisms and the potential for the identification of evolutionarily conserved oscillations in cortical DNA methylation. This is particularly relevant as high-throughput analyses provide the ability to identify results that are model specific and not necessarily translate to humans (Bolker, 2012 Nature 491, 31-33; de Souza, 2013 Nat Methods 10,288). The data presented in the submission indicates that only 10% of rhythmic mouse genes are similar to humans (Figure S2). The pattern suggests the findings are largely mouse specific. It is important to emphasize this does not lessen the value of the manuscript or the significance of the results. The paper would benefit from more detailed text on the findings (particularly in the abstract) and less on translation.

3) Is there a functional outcome of the 250 gene identified to be similar in human and mouse models? Lines 132-134. The subsequent bioinformatic analyses of cyclical changes in promoter, enhancer, gene bodies and intergenic regions is not sufficient to infer physiology or morphological consequences. Are there any other publications that could address the physiology/behavior/morphological or cognitive implications?

4) The analyses have used cortical tissue. This is slightly problematic from a circadian perspective as the cortex is a 'slave' to the main hypothalamic oscillator (i.e

suprachiasmatic nucleus). The observation of rhythmic DNA methylation in the cortex is very important and needs broader readership to ensure the maximum dissemination. It would be useful to know how the epigenetic enzyme patterns in the cortex (i.e. Figure 8a) relate to circadian expression profiles in other tissues/animals for example: Sato et al., 2017 Cell 664-677; Orozco-Solis et al., 2016 Cell Metab. 467-478; Stevenson, 2017 Gen Comp Endo. Azzi et al., 2017 Neuron. This is particularly relevant to lines 310-316.

Minor points:

The authors should indicate the logic for using heterozygous snord116 mutant mice in their analyses. This is important as there are other phenotypic effects that are likely contributing to the results.

Line 52-54: the word 'thrive' is an unusual word choice. Perhaps rephrase the sentence. Do the cortices used for mouse and human comparisons have functional homology? It is unclear why the authors have used the cortex in their analyses compared to other tissues or brain regions. A clear justification on the anatomical localization of the analyses is warranted.

Figure 7 is not presented well. Please find alternative color selection and/or use different (larger) symbols and line format.

In the pdf file available, the Table S12 TS locus and TS & PWI loci data are not formatted well.

Figure S3 panel B and the title are not clearly described and directional language should be employed.

Reviewer #2 (Remarks to the Author):

In this study, the authors demonstrate that circadian rhythmicity of CpG methylation in brain is lost in PWS mice. They then correlate these differences with gene expression, chromatin compaction, and function. The conclusions of this paper are fascinating, and relevant to a broad audience of biologists. However, to me they are severely hampered by their experimental design. I would be much more enthusiastic about this paper if I were sure that the observations were robust. I suggest how to go about this (without redoing everything) below.

1. Whole-genome bisulfite sequencing is definitely a thorough approach, but they include CpGs with only a single read. These are grouped together into methylated regions with relatively minimal criteria: CpGs must be within 300 bp of each other, and at least 10% different. Given the mapping stringency (which is not stated), what is the error rate within these included single-read samples?

2. The design of timepoints in the methylation experiments is very suboptimal, both sparse and with unequally spaced times of collection. As a result of it, they are forced to define rhythmic as "oppositely methylated DMRs at ZT0v6 and ZT6v12, with DMRs identified from ZT0v12 removed." Really, this identifies things in which ZT6 is different, a salient feature of their wildtype dataset, while carefully ignoring that many of these things are equally

different at ZT16 (in 2 out of 3 wildtype samples). Thus, they call their samples “diurnally rhythmic”, and are thereby able to claim that this is lost in PWS mice: the PWS model mice show a markedly attenuated ZT6 peak, but an increased ZT16 one.

As a result of these problems, the whole discussion of “nadir gained”, “nadir lost”, etc., is of questionable meaning, but is the best that they can do with their timeseries. Can the authors defend this?

3. The ideal thing would be to choose timepoints better designed to identify rhythmicity, but at this point I think that the comment would not be fair. I think that much of the hesitation with the design described in the previous two points would be allayed if an alternative method were used to confirm some of these...methyl-sensitive restriction site qPCR, etc...this time with a better-designed timeseries to catch rhythmic differences. The authors do examine chromatin compaction with FISH in two instances, but this is not a substitute.

4. Secondly, it is impossible to see from the data the amplitude of any of the methylation effects observed. Can the authors please 1) create a graph showing the oscillation amplitudes observed on one axis, and the number of reads at that CpG on the other, and 2) show a distribution of oscillation amplitudes for wt and pws on the same axis when the data are z-scored together. For both of these, please define oscillation as simple max/min, and not the more subtle metric used by the authors. In the first, I am hoping to see that reasonable oscillation is observed at all loci, and not just those read the least. In the second, I am hoping to see a change in the amplitude of oscillation overall, rather than a shift in phase that reads as a “loss”.

Minor comment:

l.162 “this loss of rhythmicity is not lost” could be clarified

Reviewer #3 (Remarks to the Author):

Coulson and colleagues present data from WT and Snord116+/- mice kept under LD conditions, and sacrificed at ZT0, ZT6, ZT12, and ZT16 for whole-genome bisulfite sequencing, ZT0, ZT3, ZT6, ZT9, ZT12, and ZT16 for FISH targeted toward the imprinted PWS and TS loci, and ZT6 and ZT16 for RNA seq. Light period rhythmic sites were defined as $(ZT0=ZT12) \& ZT6 <> ZT0 \& ZT6 <> ZT12$ with unadjusted $p < 0.05$. Dark period rhythmic sites were defined as $(ZT0=ZT12) \& ZT16 <> ZT0 \& ZT16 <> ZT12$ with unadjusted $p < 0.05$. They also defined a set of differentially methylated sites between Snord116+/- and WT (called PWS differentially methylated sites). They compared these to sets of sites differentially methylated in human PWS, and rhythmic in human brain, and examined intersections between sets of sites and enrichment between sets.

The reported that the set of WT light period rhythmic sites overlapped substantially with PWS differentially methylated sites, and with human rhythmic sites. Snord116+/- genotype was associated with a loss of light period rhythmic sites as defined above, and a gain of dark period rhythmic sites, with some of WT light period rhythmic sites becoming dark period

rhythmic sites in Snord116+/- . These differentially rhythmic sites were enriched for location in promoters and enhancers, and for genes (identified by RNA-seq) upregulated at ZT6. They were also enriched for pathways/phenotypes implicated in circadian rhythms and metabolics. Snord116+/- genotype was also associated with alterations in the rhythmicity of chromatin state at the imprinted PWS and TS loci.

I think the overall message - that Snord116 deletion (as a model of PWS) is associated with changes in the diurnal rhythmicity of DNA methylation as multiple loci also implicated in key PWS pathways and phenotypes, is both plausible and of considerable interest as it highlights the possibility that a critical element of PWS pathogenesis is disruption of diurnal rhythmicity at the epigenetic level. This has implications for potential epigenetic chronotherapeutics. However, I have a few important concerns about the design of the experiment (in particular the choice of sampling times) and also about the way the data were analyzed.

1) The authors used definitions of rhythmicity that were very phase specific and really would only have picked up sites whose nadirs were at ZT6 and ZT18. Indeed, even a high amplitude site whose nadir/acrophase were at ZT0 and ZT12 would not have been rhythmic by the authors' criteria. This creates a number of problems:

a) it is far from guaranteed that the bulk of rhythmic CpG sites will have acrophases/nadirs near ZT6 and/or ZT 18. Indeed, in the study cited in reference 9 of the present manuscript, the bulk of rhythmic sites had acrophases/nadirs closer to ZT0 and ZT12 than ZT6 and ZT18, and these would not have been called rhythmic by the authors' criteria

b) The ability to detect rhythmicity preferentially at very specific phases makes it very difficult to interpret a "loss" of rhythmicity, since this could reflect either a phase shift or a loss of amplitude. Even a strongly rhythmic (in WT) CpG with nadir ZT6, if phase shifted by 6 hours with no loss of amplitude, would appear completely arrhythmic by the authors' criteria. That is, if the main effect of Snord116 deletion is to cause a 6-hour phase shift, then there would be an apparent loss of rhythmicity of essentially all ZT6 nadir/acrophase sites without any loss in amplitude.

c) in light of b) above, the claim in the abstract that "of the >23,000 diurnally rhythmic CpG's identified in wild-type cortex, 97% lost rhythmic methylation in PWS cortex" is I think potentially quite misleading. The way it is written, it sounds as though PWS is associated with a dramatic loss of rhythmicity in the methylome. Yet, much of this may simply represent a phase shift. Indeed, as the authors themselves note, a substantial fraction of the sites whose rhythmicity was "lost" were simply phase shifted to the dark phase. It is distinctly possible that the remainder were simply phase shifted to intermediate phases not picked up by the authors' analysis. Put another way, based on the authors' data, one might just as easily say in the abstract that of the 13,260 dark phase rhythmic CpGs in PWS cortex, >90% are "lost" in the wild type cortex and there is therefore a defect of rhythmicity in WT compared to PWS cortex.

2) Related to the above, I think the authors' choice of ZT16 as the dark phase sampling time, resulting in uneven sampling intervals around the 24-hour clock, is also problematic. This makes it relatively more likely to pick up ZT6 (light phase) acrophase/nadir sites than ZT18 (dark phase) acrophase/nadir sites (since ZT16 is quite close to ZT12 making it harder to find sites where ZT16<>ZT12)

3) My third main concern is with respect to the issue of multiple comparisons. The authors

report that light-specific rhythmic CpGs comprises 0.31% of all CpGs. However, without knowing the null distribution for this statistic, it is difficult to know whether this is more or less than expected by chance. This issue is also true for the comparison of rhythmicity between WT and Snord116^{+/-} cortex - even in the event of pure random noise, some number of WT CpGs will meet the authors' rhythmicity criteria simply by chance alone, and by chance alone, these will be very likely not to be rhythmic in the Snord116^{+/-} cortex (and vice versa - by chance alone some number of sites will be apparently rhythmic in Snord116^{+/-}, and a large fraction of these will not be rhythmic in WT). Thus, it is critical to demonstrate that the number of rhythmic sites in each condition exceeds that expected by chance alone, and moreover, that the difference between conditions also exceeds that expected by chance alone. Given the unknown and likely complex correlations between CpG sites, this would be very hard to model parametrically. Therefore, creation of null datasets by permutation of ZTs, and then use of these null datasets to create empiric distributions of the statistics used in the paper, would provide important reassurance that the results are more than expected by chance.

We thank the reviewers for their time spent reading and commenting on our manuscript and for helpful suggestions for improvement. We appreciate the overall enthusiasm for the broad interest and impact of our findings and feel that our revised manuscript has benefitted from the addition of new data and new analyses requested by the reviewers (Figures 1, 7, S1, S2, S3, S4c, S5, S6b, S7b, and Supplementary Tables 1, 2, and 13). The revised manuscript now includes new triplicate WGBS datasets for the ZT3 and ZT9 time points. These new time points provide a more thorough sampling of light hours, where we see the greatest genotype effect, but do not change the overall conclusions of the study. A second dark time point at ZT18 has also been sampled and assayed by pyrosequencing. We have also added a new author, Yihui Zhu, who contributed to the pyrosequencing in the revised manuscript. Changes to the text are shown in blue font. Specific responses to the reviewers' critiques are below:

Reviewer 1

1) The 'Supplementary Table' attachment only has Table S12; and does not have Table S1-S11

Tables S1-S13 are included as individual tabs of one large Excel file. Perhaps these were lost in the conversion to pdf. Please see the associated xls file as a Supplemental dataset and look for individual tabs for each Supplemental table.

2) The manuscript would significantly benefit from a streamlined approach; less on the 'translational' aspects and more on the exciting findings on basic mechanisms and the potential for the identification of evolutionarily conserved oscillations in cortical DNA methylation. This is particularly relevant as high-throughput analyses provide the ability to identify results that are model specific and not necessarily translate to humans.

We appreciate this insight into the limitations of translating findings in model organisms to humans and have addressed these limitations in the Discussion section on page 7. We have also included a brief introductory statement clarifying the relevance of cross regulation between the Prader-Willi/Angelman syndrome and Temple/Kagami-Ogata syndrome loci as part of an imprinted gene network which regulates processes such as metabolism and sleep on page 2.

3) Is there a functional outcome of the 250 gene identified to be similar in human and mouse models? Are there any other publications that could address the physiology/behavior/morphological or cognitive implications?

We have provided a list of selected genes found to be rhythmic, associated with PWS, and shared between species in the new Figure S4c, with a full list in Supplementary Table 4. These genes are relevant to mTOR signaling, circadian entrainment, sleep, diabetes, and cholesterol.

4) The observation of rhythmic DNA methylation in the cortex is very important and needs broader readership to ensure the maximum dissemination. It would be useful to

know how the epigenetic enzyme patterns in the cortex (i.e. Figure 8a) relate to circadian expression profiles in other tissues/animals.

To provide perspective on the broad implications of the rhythmic expression of epigenetic enzymes, we have included a brief discussion of the diverse biological processes regulated by these patterns in various tissues and organisms in the Discussion section on page 8 of the revised manuscript.

Minor points:

The authors should indicate the logic for using heterozygous snord116 mutant mice in their analyses.

Because the Prader-Willi locus is parentally imprinted, deletions specifically of the paternal allele result in Prader-Willi syndrome, therefore a heterozygous (paternal) deletion of *Snord116* is the appropriate model for Prader-Willi syndrome. We have provided this explanation more clearly in the Introduction on page 2.

Line 52-54: the word 'thrive' is an unusual word choice

"Failure to thrive" is the standard clinical term for infants that fail to gain weight in the neonatal period, and is also a standard phenotypic characteristic of Prader-Willi syndrome infants.

Do the cortices used for mouse and human comparisons have functional homology?

While there are large differences in size and morphology between mouse and human cortices, cognitive and executive functions are common to both mouse and human cortex. We have now added a brief statement about the shared cortical functions between the two species to the Discussion on page 7.

Figure 7 is not presented well. Please find alternative color selection and/or use different (larger) symbols and line format.

We have changed the colors used in Figure 7 to improve clarity.

In the pdf file available, the Table S12 TS locus and TS & PWI loci data are not formatted well.

This is likely due to a conversion issue, as the supplementary tables were originally in excel file format. We have attached the Supplemental Tables as a separate tabbed xls file as a dataset.

Figure S3 panel B and the title are not clearly described and directional language should be employed.

This figure legend (now Figure S6b) has been revised to describe better the data being graphed.

Reviewer 2

1) Whole-genome bisulfite sequencing is definitely a thorough approach, but they include CpGs with only a single read. These are grouped together into methylated regions with relatively minimal criteria: CpGs must be within 300 bp of each other, and at least 10% different. Given the mapping stringency (which is not stated), what is the error rate within these included single-read samples?

Mapping stringency was three mismatches for reads without adapters (90 bp after quality trimming) and two mismatches for adapter trimmed reads (80 bp after quality and adapter trimming). This information has been added to our Methods section on WGBS sequencing on page 10. Although the minimum coverage requirement per sample to call differentially methylation regions was set to 1x, we have included a new analysis of rhythmicity by coverage (Supplementary Figure 3a), showing that most CpGs analyzed for rhythmicity were covered at a greater depth than the minimum threshold required, with rhythmic methylation patterns remaining fairly consistent at different coverage levels.

2) The design of timepoints in the methylation experiments is very suboptimal, both sparse and with unequally spaced times of collection.

In the revised manuscript, we have sampled two additional time points during the light phase in triplicate for each genotype by WGBS. Since the light cycle is when the phenotype of *Snord116* deletion mice is observed, it is appropriate to have time points sampled at 3 h intervals during light hours. We also have included an additional mid-dark phase ZT18 time point in triplicate for each genotype in the new pyrosequencing analyses (Supplementary Figure 3b).

3) I think that much of the hesitation with the design described in the previous two points would be allayed if an alternative method were used to confirm some of these...methyl-sensitive restriction site qPCR, etc...this time with a better-designed timeseries to catch rhythmic differences.

In the revised manuscript, we have performed pyrosequencing analyses that confirmed the rhythmicity and PWS phase shift for two light phase rhythmic DMRs observed by WGBS. Two additional light hour time points were sampled (ZT3 and ZT9) as well as a mid-dark phase time point at ZT18 (Supplementary Figure 3b).

4) Can the authors please 1) create a graph showing the oscillation amplitudes observed on one axis, and the number of reads at that CpG on the other, and 2) show a distribution of oscillation amplitudes for wt and pws on the same axis when the data are z-scored together.

In the new Supplementary Figure 3a, we have graphed oscillation amplitudes for each CpG coverage level observed. Although we required one read to cover each CpG, the majority of CpGs are covered by more than one read, and oscillations become more distinct with increasing coverage, confirming that low coverage CpGs are not driving the rhythmic pattern observed.

We have also graphed the distribution of oscillations for WT and PWS as z-scores in the new Supplementary Figure 2. Z-scores were calculated from combined WT and PWS samples to examine deviation from the total methylation distribution based on time point. WT samples showed a greater deviation from the mean during early light hours (ZT3-ZT6), representing the nadirs observed at this time point. In contrast, no large changes in distribution were observed for any single time point in PWS samples, suggesting that these nadirs were not exclusively shifted to another single time point in PWS. Oscillation amplitudes across the entire time series were also represented in combined graphs for each genotype, showing the spread of deviation from the mean overtime for WT and PWS and demonstrating a larger amplitude of oscillation in WT overall compared to PWS.

Minor point:

I.162 "this loss of rhythmicity is not lost" could be clarified.

We have clarified this statement in the text on page 4.

Reviewer 3

1a) It is far from guaranteed that the bulk of rhythmic CpG sites will have acrophases/nadirs near ZT6 and/or ZT18.

From our principal component analysis of DMRs called on the basis of genotype only (Figure 2a), we saw an effect of time between ZT0-ZT6 and ZT6-ZT12 in WT cortex, suggesting that methylation patterns at ZT6 were distinctly different from neighboring sampled time points. This rationale is now better clarified on page 4. This time effect was not observed at any other time point sampled, therefore we focused our analyses on ZT6 for identification of rhythmic CpG sites. In response to Reviewer 2, we have also now included two additional time points during light hours by WGBS in the revised Figure 1, additional dark and light time points to pyrosequencing analysis in Supplementary Figure 3b, and the Z-score analysis in Supplementary Figure 2.

1b and 1c) The ability to detect rhythmicity preferentially at very specific phases makes it very difficult to interpret a "loss" of rhythmicity, since this could reflect either a phase shift or a loss of amplitude. The claim in the abstract that "of the >23,000 diurnally rhythmic CpG's identified in wild-type cortex, 97% lost rhythmic methylation in PWS cortex" is I think potentially quite misleading. The way it is written, it sounds as though PWS is associated with a dramatic loss of rhythmicity in the methylome. Yet, much of this may simply represent a phase shift.

In the revised Figures 1, S1, S2, S3b, and S5, we have increased our sampling distribution during the light phase to include ZT3 and ZT9, to provide a higher resolution view of methylation patterns at three hour intervals. By examining the methylation pattern of rhythmic DMRs across all time points (ZT0, 3, 6, 9, 12, 16) in both WT and PWS cortex (Figure 1b), we do not see a distinct shift of identical DMRs in PWS, however we do observe a modest nadir in PWS cortex at ZT16, including DMRs which overlap those lost at ZT6 as well as unique DMRs. The conclusions in our revised abstract and Discussion are consistent with both a loss of ZT6 methylation nadirs as well as a shift of a portion of the lost nadirs to the dark phase.

2) I think the authors' choice of ZT16 as the dark phase sampling time, resulting in uneven sampling intervals around the 24-hour clock, is also problematic.

In the new Supplementary Figure 3b, we have included an additional ZT18 dark phase time point in pyrosequencing analyses. These analyses indicate that the light period from ZT3-ZT6 and the dark period from ZT12-ZT16 appear to be the critical windows for methylation changes in WT and PWS, respectively.

3) Creation of null datasets by permutation of ZTs, and then use of these null datasets to create empiric distributions of the statistics used in the paper, would provide important reassurance that the results are more than expected by chance.

We have created a null dataset by permutation of ZTs and analyzed each permutation for rhythmicity using the same methods as for the experimental groups. By Fisher's Exact test, WT and PWS rhythmicity were both significantly greater than expected by chance from the null distributions for each genotype respectively, and WT rhythmicity was significantly greater than PWS rhythmicity. This has been noted in the text on page 7, with a description of the analysis added to the Methods section under "Assignment of ZT6 diurnally rhythmic DMRs and CpGs" on page 10, and statistical data included as a new Supplemental Table 13.

Reviewers' comments:

Reviewer #1 (Remarks to the Author):

The authors have addressed each concern and minor points raised, and the manuscript has been significantly improved. It appears that one of my concerns was not clearly articulated. It is important to state which cortical region the tissue samples were derived. Was the tissue collected from the whole cortex or an anatomically localized area? (e.g. olfactory cortex, visual cortex). If the cortical sample was anatomically localized, how does the selected region relate to Prader-Willi/Angelman syndrome and Temple/Kagami-Ogata syndrome?

Reviewer #2 (Remarks to the Author):

I think that the authors have fully addressed my concerns. I support the publication of this paper.

One minor detail: in describing the new Fig S3, the authors should specify that n=number of rhythmic CpGs, not n=number of CpGs. This confused me for awhile.

Reviewer #3 (Remarks to the Author):

RE the issue of permutation testing, this issue appears to be reasonably addressed.

RE the issue of sampling times, the addition of additional sampling points in the light phase (and 1 mid-dark phase sampling point at ZT18) does partially address the issue. However, the coverage during the dark phase is still relatively sparser than during the light phase, especially for the WGBS analysis (the additional dark phase time point was for pyrosequencing only). This still limits the capacity to identify diurnally rhythmic CpG's with acrophases/nadirs at time points other than ZT6 [i.e. if one were to define a ZT3 diurnally rhythmic CpG the same way as one defined a ZT6 diurnally rhythmic CpG, one would need to compare ZT3 to ZT9 (which is now available) and ZT21 (which is not). I appreciate that the authors have presented some data to support the idea that ZT6 is where the "action" is, but the asymmetry of sampling times around the 24-hour clock still leaves the possibility that ZT6 appears to be where the action is in part because the densest sampling was centered at ZT6 +/- 6 hours (i.e. it would be harder to pick up at ZT3 acrophase/nadir CpG because of the lack of the ZT21 sampling time). The sparseness of sampling in the dark phase, while improved, also still limits the capacity to detect phase shifts in Snord vs. WT as phase shifts to phases other than ZT18/16 nadir/acrophase would still be relatively difficult to detect.

In an ideal universe, there would be symmetric samples taken every 3 (or 4) hours around the clock, but this is in likelihood simply not possible at this point without repeating the experiment. Besides, I'm not sure this issue detracts too much from what the authors did find, which is a bunch of temporally differentially methylated sites at ZT6 (vs. ZT0 and

ZT12) whose temporal patterns of methylation are different between WT and Snord116 animals, which is a very interesting finding. However, it does remain a real limitation both with regard to what they had less power to look for (i.e. rhythmic methylation sites with acrophases/nadirs at times other than ZT6) and with regard to the interpretation of loss vs. phase shift of rhythmicity. Perhaps a more thorough discussion of these limitations would help to better contextualize what they did find, and the inherent limitations of the design.

We thank the reviewers and the editor for helpful comments for the improvement of our manuscript. We have revised the text to address the suggestions raised and improve the discussion of our findings. Specific responses to the reviewers' critiques are below:

Reviewer #1 (Remarks to the Author):

The authors have addressed each concern and minor points raised, and the manuscript has been significantly improved. It appears that one of my concerns was not clearly articulated. It is important to state which cortical region the tissue samples were derived. Was the tissue collected from the whole cortex or an anatomically localized area? (e.g. olfactory cortex, visual cortex). If the cortical sample was anatomically localized, how does the selected region relate to Prader-Willi/Angelman syndrome and Temple/Kagami-Ogata syndrome?

We have specified mouse and human cortical regions sampled in the text on pages 2 and 3 as well as in the methods sections "Whole genome bisulfite sequencing (WGBS)" and the new section "Determination of human PWS DMRs" on page 10. We also include a statement that these cortical regions are relevant to the cognitive deficits observed in these disorders.

Reviewer #2 (Remarks to the Author):

I think that the authors have fully addressed my concerns. I support the publication of this paper.

One minor detail: in describing the new Fig S3, the authors should specify that n=number of rhythmic CpGs, not n=number of CpGs. This confused me for awhile.

We have clarified this point in the legend of Fig S3 on page 16.

Reviewer #3 (Remarks to the Author):

RE the issue of permutation testing, this issue appears to be reasonably addressed.

RE the issue of sampling times, the addition of additional sampling points in the light phase (and 1 mid-dark phase sampling point at ZT18) does partially address the issue. However, the coverage during the dark phase is still relatively sparser than during the light phase, especially for the WGBS analysis (the additional dark phase time point was for pyrosequencing only). This still limits the capacity to identify diurnally rhythmic CpG's with acrophases/nadirs at time points other than ZT6 [i.e. if one were to define a ZT3 diurnally rhythmic CpG the same way as one defined a ZT6 diurnally rhythmic CpG, one would need to compare ZT3 to ZT9 (which is now available) and ZT21 (which is not). I appreciate that the authors have presented some data to support the idea that ZT6 is

where the "action" is, but the asymmetry of sampling times around the 24-hour clock still leaves the possibility that ZT6 appears to be where the action is in part because the densest sampling was centered at ZT6 +/- 6 hours (i.e. it would be harder to pick up at ZT3 acrophase/nadir CpG because of the lack of the ZT21 sampling time). The sparseness of sampling in the dark phase, while improved, also still limits the capacity to detect phase shifts in Snord vs. WT as phase shifts to phases other than ZT18/16 nadir/acrophase would still be relatively difficult to detect.

In an ideal universe, there would be symmetric samples taken every 3 (or 4) hours around the clock, but this is in likelihood simply not possible at this point without repeating the experiment. Besides, I'm not sure this issue detracts too much from what the authors did find, which is a bunch of temporally differentially methylated sites at ZT6 (vs. ZT0 and ZT12) whose temporal patterns of methylation are different between WT and Snord116 animals, which is a very interesting finding. However, it does remain a real limitation both with regard to what they had less power to look for (i.e. rhythmic methylation sites with acrophases/nadirs at times other than ZT6) and with regard to the interpretation of loss vs. phase shift of rhythmicity. Perhaps a more thorough discussion of these limitations would help to better contextualize what they did find, and the inherent limitations of the design.

We agree that in an ideal universe, we would have had symmetric samples taken every 3 hours, but because of the high costs associated with sequencing three replicates of each genotype at every time point, we focused our investigation on time points covering the light hours because that is where the *Snord116* phenotype was specifically apparent in our prior metabolism and RNA-seq analyses. We have added a sentence to the introduction on page 2 addressing the light phase specific changes observed previously. We have also added a discussion of the limitations of our dark phase sampling density and the potential for future studies to the discussion section on page 8.

REVIEWERS' COMMENTS:

Reviewer #3 (Remarks to the Author):

The authors have addressed the outstanding concerns.